# Development of the Gastrointestinal Tract in Newborns as a Challenge for an Appropriate Nutrition: A Narrative Review

**DOI:** 10.3390/nu14071405

**Published:** 2022-03-28

**Authors:** Flavia Indrio, Josef Neu, Massimo Pettoello-Mantovani, Flavia Marchese, Silvia Martini, Alessia Salatto, Arianna Aceti

**Affiliations:** 1Department of Medical and Surgical Science, Pediatric Section, University of Foggia, 71122 Foggia, Italy; mpm@unifg.it (M.P.-M.); flavia.marchese@hotmal.it (F.M.); 2Department of Pediatrics, Division of Neonatology, University of Florida College of Medicine, 1600 SW Archer Road, Gainesville, FL 32610, USA; neuj@peds.ufl.edu; 3European Pediatric Association, Union of National European Pediatric Societies and Associations, 10115 Berlin, Germany; silvia.martini9@unibo.it (S.M.); arianna.aceti2@unibo.it (A.A.); 4Association pour l’Activité et la Recherche Scìentifiques, 2000 Nouchâtel, Switzerland; 5Department of Pediatrics, Scientific Institute ‘Casa Sollievo della Sofferenza’, University of Foggia, 71122 Foggia, Italy; 6Department of Medical and Surgical Sciences, University of Bologna, 40138 Bologna, Italy; 7Neonatal Intensive Care Unit, IRCCS AOU Bologna, 40138 Bologna, Italy; 8Division of Neonatology, Section of Pediatrics, Department of Translational Medical Sciences, “Federico II” University, 80138 Naples, Italy; alessiasalatto1@gmail.com

**Keywords:** gastrointestinal development, preterm infants, digestion, intestinal motility, enteric nervous system, microbiota, gut–brain axis, microbiota

## Abstract

The second and third trimesters of pregnancy are crucial for the anatomical and functional development of the gastrointestinal (GI) tract. If premature birth occurs, the immaturity of the digestive and absorptive processes and of GI motility represent a critical challenge to meet adequate nutritional needs, leading to poor extrauterine growth and to other critical complications. Knowledge of the main developmental stages of the processes involved in the digestion and absorption of proteins, carbohydrates, and lipids, as well as of the maturational phases underlying the development of GI motility, may aid clinicians to optimize the nutritional management of preterm infants. The immaturity of these GI systems and functions may negatively influence the patterns of gut colonization, predisposing to an abnormal microbiome. This, in turn, further contributes to alter the functional, immune, and neural development of the GI tract and, especially in preterm infants, has been associated with an increased risk of severe GI complications, such as necrotizing enterocolitis. Deeper understanding of the physiological colonization patterns in term and preterm infants may support the promotion of these patterns and the avoidance of microbial perturbations associated with the development of several diseases throughout life. This review aims to provide a global overview on the maturational features of the main GI functions and on their implications following preterm birth. We will particularly focus on the developmental differences in intestinal digestion and absorption functionality, motility, gut–brain axis interaction, and microbiomes.

## 1. Introduction

The maturation of the gastrointestinal (GI) system in full-term and preterm human infants is an area of great interest from both a nutritional and medical practice standpoint. Particularly in preterm infants less than 28 gestational weeks (GW), delivery constitutes a nutritional emergency in which the infant has high and difficult-to-meet nutritional needs [1]. A contributing factor to the nutritional emergency is the relatively underdeveloped GI system of premature neonates, which limits their ability to utilize enteral nutrition. The GI system of preterm infants exhibits reduced digestive and absorption capacities, prolonged gastric emptying times, and limited intestinal motility compared to term infants. [2,3]. These same limiting factors that lead to a nutritional crisis alter the preterm infant’s response to orally administered therapeutic agents [4].

The disorders of GI ontogenesis, along with other factors such as early postnatal stress, microbiota alterations induced by infections, or early antimicrobial use in the Neonatal Intensive Care Unit (NICU), result in an impaired activation of intestinal peristalsis and of the gut–brain axis [5]. Disturbances of physiological inflammatory responses further contribute to this impairment. Abnormal development of bowel function is the main determinant of food intolerance, a major problem in the NICU.

Newborns need the structural and functional maturation of the GI tract for digestion and absorption of the nutrients from colostrum and breast milk. They also need a complete development of intestinal motor function which includes suck–swallow coordination, continence of the gastroesophageal sphincter tone, adequate gastric emptying, and intestinal peristalsis. Full-term babies can acquire adequate amounts of nutrients to promote the rapid growth that occurs shortly after birth. However, half of preterm infants are delayed in reaching full enteral feeding volumes and have gastroesophageal reflux, gastric residuals, and constipation due to delayed gastric emptying, prolonged bowel transit, abdominal distension, and delayed passage of meconium, all of which are GI functions describing immaturity [6]. There are few studies available on fetal ontogenesis and early neonatal adaptation of motility and barrier functions of the human intestinal mucosa [7,8,9]. The functional components of the human GI tract do not develop simultaneously: in fact, although the anatomical differentiation of the human intestine usually occurs within 20 GW, functional maturation is postponed over time and requires organized peristalsis and coordinated sucking and swallowing, which are not defined until 29–30 and 32–34 GW, respectively [10].

In this review we will provide a global view of the basic intestinal function present in the GI tract at birth in term and preterm infants. We will pay particular attention to developmental differences in intestinal digestion and absorption functionality, motility, gut–brain axis interaction, and microbiomes.

## 2. Development of Digestion and Absorption in the Neonate

The human GI tract is an organ with one of the largest surface areas of the body. The size of the intestine exhibits an estimated one-thousand-fold increase from 5 through 40 GW [11]. Autopsy data have shown the following intestinal prenatal lengths: 125 cm by 20 GW, 200 cm by 30 GW, and 275 cm at term. This growth continues, ultimately reaching a length of approximately 575 cm by 20 years of life [12]. When including villus and microvillus structures, the surface area in the adult is estimated to be 200 m^2^. This surface is in contact with food, microbes, and other potentially antigenic components, and plays a key role in the digestive and absorptive processes [13]. The large intestine is approximately 60 cm in a full-term infant, which increases up to 150 cm by adulthood [11].

The GI tract is comprised of several structures that extend though the oropharynx, esophagus, stomach, duodenum, jejunum, ileum, cecum, and colon. Each one of these performs discrete functions critical to health. Thought to be primarily an organ of digestion, absorption, and excretion of waste, it contributes to overall health in a much broader sense [14]. In addition to digestion and absorption, the intestinal tract exerts neural, endocrine, exocrine, and immunologic functions [15,16]. The main developmental features of protein, carbohydrate, and lipid digestion are detailed in the following paragraphs and summarized in Figure 1.

### 2.1. Proteins

Protein sources for term and preterm infants are comprised of two major components, whey and casein. Digestion of these proteins is initiated in the stomach, where parietal cells secrete hydrochloric acid which, in combination with activation of the proenzyme pepsinogen, denatures some of these proteins. Parietal cells are present by the late first trimester of pregnancy, and they actively secrete gastric acid by the second trimester. [17,18]. After preterm birth, gastric acid production is limited in comparison to infants born at 40 GW. During the first two months after birth, gastric acid production doubles [19]. The proenzyme pepsinogen can be detected in the fetal stomach by 17–18 GW. After birth, pepsin activity is in line with the infant’s maturity [20].

Other major enzymes are produced by the pancreas during this time. These include trypsinogen, chymotrypsinogen, and carboxypeptidase, all of which are zymogens that require activation by enterokinase, which is produced by the upper intestinal epithelial cells. By 24 GW, enterokinase is active at 25% of the level found in older infants [21]. These enzymes break large protein molecules into oligopeptides, dipeptides and single amino acids. As these molecules move distally, the absorptive process into the small intestinal epithelial cells occurs. This is accomplished by several transporter mechanisms. There are limitations of these processes in preterm infants. [22].

This leads to the question of whether hydrolyzed protein formulas are better tolerated, thereby providing enhanced nutrition for these preterm infants. One study demonstrated that, when compared to infants receiving standard formula, infants fed hydrolyzed protein formula had better feeding tolerance and a shorter time in achieving full feeds [23].

### 2.2. Carbohydrates

Salivary and pancreatic amylases are involved in the luminal digestion of complex sugars. Digestion into oligosaccharides is then complemented by absorptive hydrolysis into monosaccharides at the epithelial brush border by enzymes that include lactase, sucrase, maltase, isomaltase, and glucoamylase. Sucrase, maltase, and isomaltase have been shown to be fully active in preterm infants. Lactase activity, which hydrolyzes lactose, which is the most abundant disaccharide, into glucose and galactose, is low in preterm infants. [21].

Salivary amylase, which digests complex carbohydrates of 18–29 glucose units, is an initiator of carbohydrate digestion [24]. Pancreatic amylase, the secretion of which is limited, not reaching adult levels until about 2 years of age, is responsible for most initial carbohydrate hydrolysis. [25,26]. Due to suck–swallow incoordination, enteral feeding in many preterm infants necessitates the use of feeding tubes. These bypass the oral cavity, hence also bypassing the activity of salivary amylase.

Undigested carbohydrates pass into the distal intestine. Here, microbial fermentation results in the production of short chain fatty acids (SCFAs), which subsequently are absorbed. These can be utilized as energy sources, but butyrate has been also shown to serve as an important fuel for colonocytes as well as having properties that alter proliferation, differentiation, and turnover of colonic cells. [27,28].

Lactose is especially interesting from this perspective. Being the primary carbohydrate source in milk, lactose intolerance rarely occurs in the preterm population. Unabsorbed lactose conversion to SCFAs provides an efficient salvage pathway. One study suggests that lactase activity can be induced in preterm infants with enteral feedings of human milk [29].

Most lactase activity is found at the middle to the top of the villus, while sucrase, maltase and glucoamylase are found at the mid-villus [30]. During time of intestinal mucosal injury, the lactase-rich cells at the villus tip are the first to be injured and the last to be fully restored through the process of cell migration from crypt to the villus tip.

Insulin plays an important role in intestinal maturation and may also contribute to improve lactase activity following preterm birth. Amniotic fluid contains carbohydrates, protein, fat, electrolytes, immunoglobulins, and growth factors. Insulin is present in the amniotic fluid (up to 20 μU/mL) and after skin keratinization completed (around 26 weeks’ gestation), amniotic fluid is the main source of insulin exposure to the GI tract and plays a role in development. Preterm birth abruptly interrupts these important fetal life processes and the local insulin exposure of the GI tract with detrimental consequences [31].

Oral insulin administration in preterm infants up to 28 days after delivery has proved effective in doubling the lactase activity [32].

### 2.3. Lipids

Triglycerides constitute half of the non-protein energy content in human milk and formula [16]. The digestion of triglycerides is initiated by bile acid micellar emulsification, which produces smaller droplets of triglycerides. This results in greater surface area for interaction with lipases, which hydrolyze long chain triglyceride into monoglycerides and free fatty acids, which in turn are absorbed into the small intestinal epithelium. Pancreatic lipase is one of the most important of these lipases.

The absorbed free fatty acids and 2-monoglycerides are re-esterified within the epithelial cell and subsequently converted to chylomicrons, which are transported from the basal region of the epithelial cell via the thoracic duct, from which they enter the blood circulation.

The digestion and absorption of medium chain triglycerides (MCTs) is less complex than that of long chain triglycerides. They do not require bile acid emulsification, are absorbed into the enterocyte without re-esterification and travel directly into the portal venous system. MCTs are frequently recommended for patients with lymphatic obstructions [33].

Different sources of lipase include lingual, gastric, pancreatic, and epithelial cells. Human milk contains a lipase termed bile-salt stimulated lipase. After activation in the small intestine in the presence of bile acids, it facilitates long-chain triglyceride digestion. Lingual lipase is secreted from glands at the base of the tongue; its activity is lower at birth in infants at 26 weeks’ gestation, peaks at 30–32 weeks, and declines near term [34,35]. Pancreatic lipase insufficiency is more prevalent in preterm infants when compared to older children. Adult levels are often not reached until 6 months after birth [36].

Another limitation of fat digestion relates to bile acids, which are synthesized and excreted from the liver via the biliary system at relatively low levels in very low birth weight (VLBW) preterm infants when compared to term infants. Bile ileal reabsorption is also lower in preterm infants, which results in less efficient lipid digestion when compared to term infants [37,38].

Essential fatty acids cannot be produced by the body and become deficient when not supplied in adequate quantities in the diet. To prevent deficiency, adequate dietary intake of linoleic and linolenic fatty acids can prevent deficiency. Furthermore, linoleic, and linolenic acids (18-carbons chain length) undergo desaturation and elongation enzymatic processes into long-chain polyunsaturated fatty acids (LCPUFAs) with more than 20-carbons. Required for formation of eicosanoids and central nervous system structures [39], it is concerning that preterm infants may not have the enzymatic capability to undergo these conversions. Many very preterm infants are provided formula that are not supplemented with these LCPUFAs, and even those receiving human milk may not receive adequate quantities of these important lipids [40].

## 3. The Development of Intestinal Motility

Intestinal motility includes coordinated movements of the GI tract such as mixing, propagation of motor activities, and receptive relaxation. These movements are regulated by multiple control systems including extrinsic neurons, intrinsic neurons (the enteric nervous system, ENS), interstitial cells of Cajal (ICCs), cells expressing platelet-derived growth factor receptor alpha (PDGFRα), and myogenic mechanisms, which can all operate simultaneously [41,42]. The principal maturational features of intestinal motility are detailed in the following paragraphs and summarized in Figure 1.

### 3.1. Anatomy and Physiology of Gastrointestinal Motility

The movements of the GI tract are different for each region. They also show marked differences in relation to prematurity [43,44].

In the literature there are several studies on GI motility in the preterm newborns compared to full term newborns, characterized by impaired gastric electrical activity and gastric emptying. Gastric motility is poorly organized in preterm infants and fundal and antral pressure waves increase with increasing gestational age (GA) [9,45,46]. In particular, this is relevant for preterm newborns of 28–32 weeks GA, while preterm infants of 32 to 36 weeks GA show a pattern of gastric electrical activity and emptying much closer to that of full-term newborns.

Gastric motility is ruled by myoelectrical activity that consists of rhythmically occurring transmembrane potential variations, called slow waves, that are independent of the presence of motor activity [47]. These waves that slowly travel through the GI tract are defined motor migratory complexes (MMCs) and occur every 2–4 h with the purpose of clearing the tract of undigested material, mucus, and debris [48]. MMCs initially occur due to vagal input and the release of motilin in the duodenum. Their propagation, on the other hand, is coordinated by enteric neurons. MMCs occur in humans only during periods of fasting and in the small intestine as opposed to other species. Phasic contraction is related to a different kind of electrical activity, called electrical response activity, characterized by spikes superimposed on gastric slow waves. The origin of electrical activity is found on the greater curve, approximately in the proximal corpus, from where the slow waves propagate towards the pylorus [48,49].

Different motor patterns occur in the proximal and distal stomach. In the proximal stomach, receptive relaxation and accommodation occur, which are both mediated by neurons in the brainstem via vagal reflexes. The distal stomach exhibits different motor patterns during feeding and fasting. In the feeding state, the distal stomach grinds and mixes. Extrinsic neurons are not essential for this contractile activity, but it can be modulated by vagal pathways.

Intestinal motor activity is also correlated with gestational age and motor development, which is defined by phases I, II, and III of the migration of the motor complex between 37 GW and term age [50]. In particular, the frequency of duodenal contractions, the number of duodenal contractions per impulse, and the peak intraluminal pressure of duodenal motility in preterm versus full-term infants are different [46]. Clustered phasic contractions are more frequent but of shorter duration and lower amplitude; duodenal clusters are less common and antro-duodenal coordination is more immature in preterm infants [51].

A role of insulin on gut motility has also been supported by Shulman et al., who demonstrated significantly reduced gastric residuals within the first month in preterm infants treated with oral insulin compared to untreated peers [32].

### 3.2. The Enteric Nervous System

By 12 GW, the fetal colon has a dense neuronal network in the myenteric plexus with expression of excitatory neurotransmitter and synaptic markers. Instead, the markers of inhibitory neurotransmitters appear no earlier than 14 GW. However, electrical train stimulation of internodal strands did not evoke activity in the ENS of 12- or 14-GW tissues [52].

Onset of evoked electrical activity in the human fetal ENS appears at approximately 16 GW. Such activity appears to coincide with increases in gene expression of various ion channels known to modulate enteric action potentials. The temporal development of several neural subtypes and enteric glia occurs between 12 and 16 GW [53,54]. The ENS consists of three parts: enteric neurons, enteric glial cells (EGCs), and ICCs. They form two major ganglionated structures and functional subunits—submucosal plexus (or Meissner’s plexus) and myenteric plexus (or Auerbach’s plexus). Meissner’s plexus is located in the connective tissue of submucosa, and innervates muscularis mucosae, intestinal neuroendocrine cells, glandular epithelium and submucosal blood vessels, while Auerbach’s plexus is located between the circular and longitudinal muscle layers and is associated with the contractility of the circular and longitudinal muscles [55,56].

Studies carried out on intestinal samples of mice show that ENS originates mainly from the cells of the vagal neural crest called neural crest precursor cells (NCC) that enter the foregut at 9 embryonic days and then migrate rostro caudally and colonize the entire intestine by 14 embryonic days [57,58,59].

Cdh19 is a direct target of Sox10 during early sacral NCC migration towards the hindgut and forms cadherin–catenin complexes which interact with the cytoskeleton in migrating cells [60]. The migration phase is followed by the proliferation of NCCs which will form millions of ENS cells. The NCCs then assemble into groups (myenteric ganglia or submucosal ganglia) and then differentiate into a range of enteric neurons and glia cells and form complex neuronal networks necessary for controlled intestinal activity [61,62,63].

Diverse populations of fibroblast-like interstitial cells are present in the adult gut. Loss or dysfunction of these cells has been linked to a wide variety of GI disorders. This broad group of cells comprises various subpopulations of Kit-positive ICCs and fibroblast-like cells expressing PDGFRα. ICCs located at the level of the myenteric plexus (ICC-MY) mediate slow waves, the electrical events that time the occurrence of phasic contractions [64,65], while evidence is accumulating that ICC and PDGFRα-expressing cells located within and surrounding GI muscle bundles serve as intermediaries in both excitatory and inhibitory neuromuscular transmission.

Unlike enteric neurons and glial cells, ICCs do not arise from the neural crest, but have an embryonic development prior to the arrival of neural crest cells in the intestine. Several studies suggest that smooth muscle cells and ICCs are derived from a common mesenchymal precursor following an epithelial–mesenchymal transition process (EMT). Both ICC and CD34+ fibroblast-like cells derive from the coelomic epithelium, most likely from a common ancestor expressing the chlorine channel anoctamin-1 and smooth muscle actin alpha. Differentiation to the ICC phenotype during embryogenesis depends on cell signaling via the receptor tyrosine kinase, Kit [66,67,68,69]. In recent studies on ontogeny of gastric electrical activity, myogenic control was found to be immature at birth and suggested a process of development from 1 week to 6 months.

### 3.3. The Microbiota-Gut–Brain Axis

During the early postnatal period, infants undergo consistent gut development, which is parallel and interdependent, even if not always synchronous, with brain development. In the first years of life, the gut experiences huge changes of the resident microbiota and a substantial maturation of both the enterocytes and of the ENS. Meanwhile, the brain and the central nervous system (CNS) grow rapidly, both in terms of brain volume and neural function.

The interdependency of the two developmental processes is mediated through the so-called gut–brain axis, which constitutes a bidirectional communication between the gut and the brain, made of several interrelated components [70]. In recent years, the role of microbiota in the gut–brain axis has been studied extensively, so that the GBA is now commonly referred to as the microbiota-GBA (MGBA) [71]. In this context, gut microbiota alterations have been directly linked to physical and mental health through a series of mechanisms which are yet to be fully elucidated [72].

The connection between gut and brain during early development involves both top-down and bottom-up mechanisms, with signals from the brain modulating GI functions, and GI-derived molecules influencing brain processes through different pathways. Even if the features of the MGBA in health and disease have been thoroughly described, the knowledge of the exact mechanisms through which the gut and the brain communicate in young infants is, at present, limited.

The mechanisms involved in the interplay between the brain and the gut include neural, endocrine, immune, and metabolic mediators [73].

Neural connections include the CNS, the ENS, and the autonomic nervous system (ANS). The ENS receives inputs from the brain and, in turn, provides ascending information through neural circuits. The ANS comprises sympathetic and parasympathetic nerves: the sympathetic system exerts mainly an inhibitory influence on the gut, while the vagus nerve (VN) appears to be able to sense hormones, cytokines, and metabolites from the GI tract, leading to afferent signals to the brain. The development and myelination of the VN is not complete at birth and continues until adolescence, with a peak in the myelination rate observed in the first months of life, when consumption of human milk, and human-milk-derived bioactive components, is the highest [74]. The ANS is connected to the limbic system of the brain, whose main components are the hippocampus, the amygdala, and the limbic cortex, and which is responsible for a variety of brain processes in health and disease [73].

As for humoral components of the MGBA, they mainly consist in the hypothalamic–pituitary–adrenal axis (HPA), the enteroendocrine system and the immune system. Stress responses are modulated through the HPA, via the release of corticosterone, adrenaline, and noradrenaline. The enteroendocrine cells (EECs) in the gut produce GI hormones such as ghrelin, glucagon-like peptide (GLP-1), cholecystokinin, and peptide YY (PYY); these hormones regulate food intake and satiety and are also involved in the regulation of emotions and mood [70].

The resident immune cells in the brain, known as the microglia, act as a neuroimmune component of the MGBA, by finely tuning neurogenesis and synaptogenesis. The maturation and function of the microglia appear to be related to changes in the gut microbiota, and also influenced by SCFAs [75], which are end-products of bacterial fermentation of dietary fibers and resistant starch in the colon. Furthermore, immune signaling molecules, whose production is also driven by gut microbiota, can play a role in the MGBA, by binding to VN receptors or by crossing the blood–brain barrier. In this respect, recent studies have suggested a potential role of bacterial peptidoglycans (PGNs), which are components of the bacterial cell wall, released not only by exogenous bacteria, but also by the resident gut microbiota. It has been proposed that PGNs could enter the systemic circulation and reach the developing brain, where PGN sensing molecules are expressed abundantly during specific time windows of perinatal development, thus potentially exerting a direct effect on brain function and development [76].

Finally, a central role in the MGBA is played by metabolic mediators, including tryptophan metabolites such as serotonin (5-hydroxytryptamine, 5-HT), melatonin, SCFAs, and other neurotransmitters. 5-HT is involved in most branches of the MGBA; for instance, it acts as a neurotransmitter both in the CNS and ENS, and 5-HT receptors have a critical function in the HPA [77]. The production of host 5-HT in the gut is regulated by microbiota, as indigenous spore-forming bacteria, derived from mouse and human microbiota, have been shown to promote 5-HT biosynthesis from enterochromaffin cells in the colon, thus impacting GI motility and homeostasis [78]. SCFAs (acetate, propionate, and butyrate) are thought to influence gut–brain communication through different potential pathways, either directly or indirectly. Once produced by colonic fermentation, SCFAs can exert a direct effect on the intestinal mucosal immunity and can modulate the integrity and function of the gut barrier. Furthermore, by interacting with the EECs, SCFAs promote a direct gut–brain signaling through the secretion of gut hormones, such as GLP-1 and PYY, and other metabolic mediators such as γ-aminobutyric acid and 5-HT. In addition, they can induce systemic inflammation through differentiation of T regulatory cells and secretion of interleukins and can probably also act centrally in the CNS by modulating neuroinflammation [79].

There are several factors which could have a specific impact on the MGBA development in early life; one of the most relevant might be prematurity, as preterm birth interrupts the physiological growth and development of both the GI tract and the nervous system and leads to a certain degree of microbial dysbiosis. Despite that, at present there are no studies specifically designed to describe the unique features of the MGBA in preterm infants [80]. In addition, nutrition during sensitive developmental time windows is thought to have a major impact on the microbiota-gut–brain crosstalk [81,82], either through an effect of single nutrients (e.g., milk fat globule membranes [83], human milk oligosaccharides [84], or through the well-known benefits of exclusive human milk feeding compared to other feeding sources [85].

Most studies aimed at describing the developmental-related characteristics of the MGBA have been carried out using rodent models, even if gut and brain development in rodents do not exactly reproduce human developmental patterns [86]. For this reason, future preclinical research in the field of MGBA should be hopefully directed towards the use of animal models with more similar anatomy and physiology to that of humans (e.g., piglets). These models would allow a better understanding of the MGBA during early development and would pave the way for translating preclinical research into clinically relevant observations.

Beyond the contribution of studies based on animal models, the plausibility of the microbiota-gut–brain interplay also during early development is supported by clinical observations: so far, a limited number of studies has shown a correlation between specific microbial features in the gut during the early postnatal period and neurodevelopment, either in term or preterm infants [87,88,89,90,91].

The potential link between microbiota and neurodevelopment is particularly intriguing for preterm infants, as prematurity itself exposes them to both early dysbiosis and intrinsic risk of altered development. It has been suggested that the interplay between early dysbiosis and the gut–brain axis might lay the foundations for the altered neurodevelopment which is often observed in preterm infants experiencing necrotizing enterocolitis (NEC) [92]. Furthermore, a potential role of early alterations of the GMBA in later neurodevelopment has also been proposed for preterm infants not necessarily experiencing major neonatal complications such as NEC. For instance, in the French EPIFLORE population study, gut microbiota features in the first month of life in very preterm infants were associated with 2 year outcomes, even after adjustment for confounders [88]. Similarly, in a small cohort of very low birth weight infants, a relationship between specific microbial characteristics during the first months of life (i.e., relative abundance of Bifidobacteria) and impaired neurodevelopment at 24 months corrected age was suggested [91].

## 4. Development of the Gut Microbiome

The intestinal microbiota is important in human growth and development. The human gut is home to trillions of microbial cells that bind in a symbiotic relationship to the host playing a vital role in both health and disease [93,94].

The relationship between microbiome and host is dynamic and many factors (such as genetic background, maternal diet prior to and during pregnancy, maternal microbiome, mode of delivery, gestational age, perinatal stress, infections during pregnancy and during the perinatal or neonatal period, feed type of neonates and especially of preterm infants, and other environmental factors such as temperature, humidity, pH, and oxygen levels in the tissues type) modulate the infants’ microbiome constitution and its dynamic relationship with the host, which can be transformed from ‘synbiotic’ to ‘dysbiotic’ and life-threatening [95,96].

Of note, as illustrated in a recent narrative review, over the past decade several studies conducted on meconium have supported the fact that the fetus could be exposed to microbiota in utero, though previously the amniotic fluid and the placenta had been thought to be sterile [97]. Placental colonization through a hematic route has been hypothesized as possibly responsible for the fetal colonization before birth [98].

Advances in high-throughput nucleotide sequencing technologies have enhanced gut microbiota research by investigating how bacterial communities develop over time, what factors affect them, and how they affect health and disease [99].

### 4.1. Gut Microbiome in Term Infants

The microbiota performs fundamental functions in human health, especially in childhood. The first year of life is considered a crucial window for any manipulation of the intestinal microbiota in order to promote proper development of the immune system and human well-being [100]. Human first exposure to these microorganisms occurs at birth: once the baby is out of the womb, the microorganisms enter and settle in the intestine.

The TEDDY study, which describes the importance of the intestinal microbiota in the first years of life, demonstrated how the development of the microbiota in a full-term infant is influenced by breastfeeding, the type of birth, the geographical location, and the presence of siblings or animals in the home [101].

Other studies, on the other hand, are focusing on the impact that maternal factors, antibiotic use, host genetics, ethnicity, and even COVID-19 infection can have on microbiota development [102,103,104]. Therefore, alterations of the intestinal ecosystem, known as dysbiosis, during this critical period of development are often associated with various kinds of pathologies.

### 4.2. Gut Microbiome in Preterm Infants

Prematurity at birth is a determining factor in the development of the intestinal microbiota [105]. The premature baby, after birth, is admitted to NICU and thus exposed more to hospital-acquired bacteria than to familiar bacteria [106]. Furthermore, the clinical practices to which these small patients are exposed, e.g., ventilation, artificial feeding, and antibiotic reception, alter the normal development of the microbiota. Moreover, pre-term infants have immature GI tracts and naïve immune systems [107].

Several studies have shown how this immaturity and diversity of the microbiota affects the development of intestinal function and anatomy of intestinal villi, crypts and of the intestinal barrier [108], causing several serious disorders in the premature newborn such as late-onset sepsis (LOS) and NEC [109]. Substantial efforts have been made over the past few decades to define the microbial characteristics of NEC and LOS, and compelling data have associated both conditions with intestinal dysbiosis.

Human milk has a protective role against both NEC and LOS as it confers immunological benefits to the baby and provides nutrient-specialized microbes that likewise enrich the baby’s immune system [110]. Specifically, human milk provides nutrients such as oligosaccharides, essential for the growth of bifidobacterial-type bacteria [111,112]. Bifidobacteria have been shown to possess immunomodulatory and anti-inflammatory properties by strengthening the intestinal barrier and reducing the risk of NEC [110]. Premature babies have low levels of these bacterial species and are more exposed to the risk of NEC. Safety and efficacy of probiotic administration to modulate early gut microbiome in preterm neonates is being investigated [113]. Gaining a deeper understanding of the microbiome using relevant methodology will enable the exploration of new targeted therapeutic strategies in preterm disease as well as in a range of other pathologies.

## 5. Conclusions

The intestine represents a vast interface between our internal and external environments. The second and third trimesters of pregnancy are crucial to the anatomical and functional development of the GI tract and the related digestive functions. Following premature birth, the immaturity of the digestive and absorptive processes and of the GI motility represent a critical challenge to the establishment of adequate enteral intakes in the preterm population and are often involved in the development of premature-related GI complications, ranging from food intolerance to NEC. In the context of gut immaturity, the choice of the optimal source of nutrition for preterm infants is fundamental: keeping mother’s own milk in mind as the magic bullet for preterm infants’ nutrition, future research should continue to address the potential role for mother’s own milk substitutes, such as donor milk and formula, for preterm infants. Specifically, the potential effects of donor milk treatments such as freezing and pasteurization on milk bioactive properties should be further investigated, while infant formula research should be oriented at developing a milk substitute which would be nutritionally adequate and, at the same time, easy to tolerate and digest for these immature infants.

Moreover, there is much evidence that microbes residing in the intestinal tract play important roles in the ontogenesis of the immune system and interact with the gut and central nervous system. The interplay between GI development and microbial colonization is dual: while the immature anatomical and functional features of the preterm gut predispose to an abnormal intestinal microbial colonization, the latter may interfere with the functional, immune, and neuronal development of the GI tract. It should be constantly kept in mind that the perinatal period most likely corresponds to a critical moment in which the “set points” are impressed; it is therefore necessary to learn more about normal and healthy colonization patterns in infants to promote these patterns and avoid perturbations that result in disease throughout life.

## Figures and Tables

**Figure 1 nutrients-14-01405-f001:**
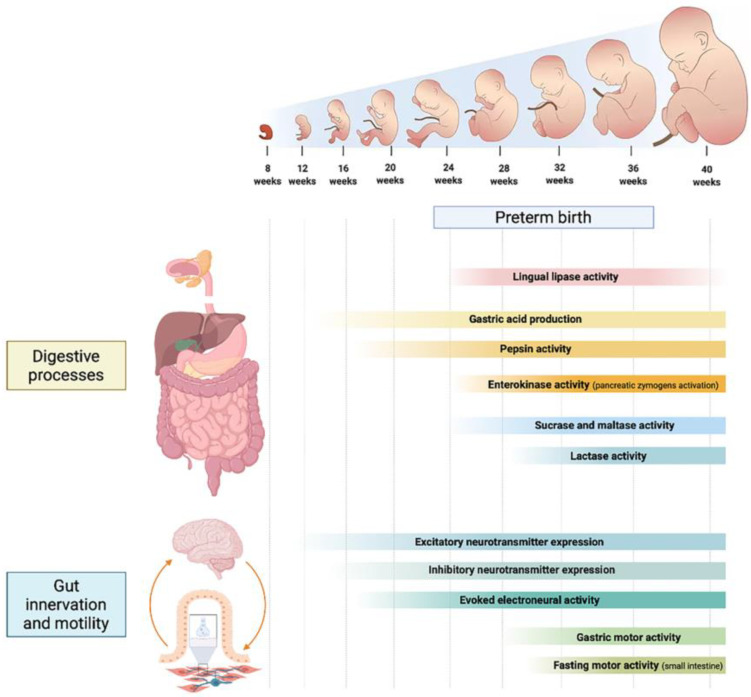
Developmental phases of the main digestive processes, gastrointestinal innervation and motility patterns throughout different gestational weeks.

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
