# Peer review of "Development of the Gastrointestinal Tract in Newborns as a Challenge for an Appropriate Nutrition: A Narrative Review"

_nutrients, 2022, doi:10.3390/nu14071405_

Round 1

Reviewer 1 Report

In this narrative review, the authors described the development of the gastrointestinal tract in newborn.

This is very interesting topics for neonatologist. There are important information for the development of the gastrointestinal tract from fetal through infant life

First in the title of manuscript should mention that this is a narrative review.

Regarding the manuscript title authors mention “DEVELOPMENT OF THE GASTROINTESTINAL TRACT IN NEWBORN: CHALLENGE FOR AN APPROPRIATE NUTRITION.” It well be better before the conclusion authors to indicate that: What is needed to move the field forward; Where appropriate, present the most important points of relevance in current clinical practice; Which areas they feel would repay further research

In the “4. Development of the gut microbiome “In this section, the authors should state that:

The relationship between microbiome and host is dynamic and many factor (such as genetic background, maternal diet prior to and during pregnancy, maternal microbiome, , mode of delivery, gestational age, perinatal stress, infections during pregnancy and during the perinatal or neonatal period, feed type of neonates and especially of preterms, and other environmental factors such as temperature, humidity, pH and oxygen levels in the tissues type) modulate the infants’ microbiome constitution and finally this dynamic relationship, which can be transformed from ‘symbiotic’ to ‘dysbiotic’ and life-threatening.

These data were published in two recent reviews. [Children 2022, 9, 154. https://doi.org/10.3390/children9020154; Diagnostics 2022, 12, 214. https://doi.org/10.3390/diagnostics12010214 ]

Author Response

Reviewer #1

In this narrative review, the authors described the development of the gastrointestinal tract in newborn. This is very interesting topics for neonatologist. There are important information for the development of the gastrointestinal tract from fetal through infant life

First, in the title of manuscript should mention that this is a narrative review.

The review title has been modified accordingly.

Regarding the manuscript title authors mention “DEVELOPMENT OF THE GASTROINTESTINAL TRACT IN NEWBORN: CHALLENGE FOR AN APPROPRIATE NUTRITION”. It well be better before the conclusion authors to indicate that: What is needed to move the field forward; where appropriate, present the most important points of relevance in current clinical practice; which areas they feel would repay further research.

The relevance of gut immaturity for preterm infants’ nutrition has been further detailed in the context of the conclusion section. In particular, the following paragraph has been added:

“In the context of gut immaturity, the choice of the optimal source of nutrition for preterm infants is fundamental: keeping own mother’s milk in mind as the magic bullet for preterm infants’ nutrition, future research should continue to address the potential role for mother’s own milk substitutes, such as donor milk and formula, for preterm infants. Specifically, the potential effects of donor milk treatments such as freezing and pasteurization on milk bioactive properties should be further investigated, while infant formula research should be oriented at developing a milk substitute which would be nutritionally adequate and, at the same time, easy to tolerate and digest for these immature infants”.

In the “4. Development of the gut microbiome “In this section, the authors should state that:

The relationship between microbiome and host is dynamic andmany factor (such as genetic background, maternal diet prior to and during pregnancy, maternal microbiome, , mode of delivery, gestational age, perinatal stress, infections during pregnancy and during the perinatal or neonatal period, feed type of neonates and especially of preterms, and other environmental factors such as temperature, humidity, pH and oxygen levels in the tissues type) modulate the infants’ microbiome constitution and finally this dynamic relationship, which can be transformed from ‘symbiotic’ to ‘dysbiotic’ and life-threatening. These data were published in two recent reviews.

[Children2022, 9, 154. https://doi.org/10.3390/children9020154;Diagnostics 2022, 12, 214.https://doi.org/10.3390/diagnostics12010214 ]

This section has been modified according to the reviewer indication, including the two mentioned references.

Reviewer #1

In this narrative review, the authors described the development of the gastrointestinal tract in newborn. This is very interesting topics for neonatologist. There are important information for the development of the gastrointestinal tract from fetal through infant life

First, in the title of manuscript should mention that this is a narrative review.

The review title has been modified accordingly.

Regarding the manuscript title authors mention “DEVELOPMENT OF THE GASTROINTESTINAL TRACT IN NEWBORN: CHALLENGE FOR AN APPROPRIATE NUTRITION”. It well be better before the conclusion authors to indicate that: What is needed to move the field forward; where appropriate, present the most important points of relevance in current clinical practice; which areas they feel would repay further research.

The relevance of gut immaturity for preterm infants’ nutrition has been further detailed in the context of the conclusion section. In particular, the following paragraph has been added:

“In the context of gut immaturity, the choice of the optimal source of nutrition for preterm infants is fundamental: keeping own mother’s milk in mind as the magic bullet for preterm infants’ nutrition, future research should continue to address the potential role for mother’s own milk substitutes, such as donor milk and formula, for preterm infants. Specifically, the potential effects of donor milk treatments such as freezing and pasteurization on milk bioactive properties should be further investigated, while infant formula research should be oriented at developing a milk substitute which would be nutritionally adequate and, at the same time, easy to tolerate and digest for these immature infants”.

In the “4. Development of the gut microbiome “In this section, the authors should state that:

The relationship between microbiome and host is dynamic andmany factor (such as genetic background, maternal diet prior to and during pregnancy, maternal microbiome, , mode of delivery, gestational age, perinatal stress, infections during pregnancy and during the perinatal or neonatal period, feed type of neonates and especially of preterms, and other environmental factors such as temperature, humidity, pH and oxygen levels in the tissues type) modulate the infants’ microbiome constitution and finally this dynamic relationship, which can be transformed from ‘symbiotic’ to ‘dysbiotic’ and life-threatening. These data were published in two recent reviews.

[Children2022, 9, 154. https://doi.org/10.3390/children9020154;Diagnostics 2022, 12, 214.https://doi.org/10.3390/diagnostics12010214 ]

This section has been modified according to the reviewer indication, including the two mentioned references.

Reviewer 2 Report

The authors aimed to provide a global overview on the maturational features of the main GI functions and on their implications following preterm birth. They particularly focussed on the developmental differences in intestinal digestion and absorption functionality, motility, gut brain axis interaction and microbiome.

This is a well-done review on the topic of “DEVELOPMENT OF THE GASTROINTESTINAL TRACT IN NEWBORN: CHALLENGE FOR AN APPROPRIATE NUTRITION.”

There are no major concerns

Minor comments:

Please cite Kurath-Koller et al. Hospital Regimens Including Probiotics Guide the Individual Development of the Gut Microbiome of Very Low Birth Weight Infants in the First Two Weeks of Life. Nutrients 2021

Author Response

Reviewer #2

The authors aimed to provide a global overview on the maturational features of the main GI functions and on their implications following preterm birth. They particularly focused on the developmental differences in intestinal digestion and absorption functionality, motility, gut brain axis interaction and microbiome.

This is a well-done review on the topic of “DEVELOPMENT OFTHE GASTROINTESTINAL TRACT IN NEWBORN:CHALLENGE FOR AN APPROPRIATE NUTRITION.”

There are no major concerns

Minor comments:

Please cite Kurath-Koller et al. Hospital Regimens Including Probiotics Guide the Individual Development of the Gut Microbiome of Very Low Birth Weight Infants in the First Two Weeks of Life. Nutrients 2021

The reference has now been included in the microbiota section.

Reviewer 3 Report

Very nice and comprehensive review

Author Response

Reviewer #3

Very nice and comprehensive review

Reviewer 4 Report

Flavia Indrio et al sent an article entitled „DEVELOPMENT OF THE GASTROINTESTINAL TRACT IN NEWBORN: CHALLENGE FOR AN APPROPRIATE NUTRITION” for publication to Nutrients.

This paper provides an overview on the maturational features of the main gastrointestinal tract (GI) functions, their implications following premature delivery focusing on the developmental differences in intestinal digestion and absorption functionality, motility, gut-brain axis interaction and microbiome.

Adequate nutrition of preterm infants still remains a challenge with regard to achieving a sufficient amount of bioactive substances assuring linear growth and neurocognitive development. During fetal life from the maternal circulation and amniotic fluid hormones are available until the development of fetal endocrine system will be completed. Preterm birth disrupts the maternal origin, the breast milk (BM) is the only source of maternal hormones. Feeding preterm infants, the own mother’s milk is the first choice, if it is not available, donor milk (DM) is recommended. It is important to note that milk composition of mothers who gave birth to preterm infants differs from breast milk produced for term infants. It was proven also that human milk reduces the incidence of necrotizing enterocolitis and several morbidities compared to formula fed infants. For microbiological safety DM milk is exposed to Holder pasteurization (HoP), however, this procedure may influence the levels of bioactive compounds in human milk.

Suggestions:

  1. In the present paper the effect of hormones and bioactive properties on anatomical and functional development of the GI tract is not emphasized to the extent of its importance.

Insulin plays an important role in intestinal maturation, also improves feed tolerance and has an impact on microbiome development.

Amniotic fluid contains carbohydrates, protein, fat, electrolytes, immunoglobulins, and growth factors. Insulin is present in the amniotic fluid (up to 20 µU/ml) and after skin keratinization completed (around 26th gestational week), amniotic fluid is the main source of insulin exposure to the GI tract and plays a role in development. Preterm birth abruptly interrupts these important fetal life processes and the local insulin exposure of the GI tract with detrimental consequences.

(J Neu, Nan Li:

The Neonatal Gastrointestinal Tract: Developmental Anatomy, Physiology, and Clinical Implications

Neoreviews (2003) 4 (1): e7–e13.

https://doi.org/10.1542/neo.4-1-e7.)

Shehadeh N et al showed that the insulin concentration of human milk after full term delivery (60.2 (41.1) mU/ml) is significantly higher than that of cow’s milk (16.3 (6.0) mU/ml), and that insulin is barely detectable in infant formulas.

(N Shehadeh, E Khaesh-Goldberg, R Shamir, R Perlman, P Sujov, A Tamir, I R Makhoul:

Insulin in human milk: postpartum changes and effect of gestational age

Arch Dis Child Fetal Neonatal Ed 2003;88:F214–F216)

Human milk insulin concentrations in mothers of preterm infants were not significantly different from those of full-term infants, on either day 3 or day 10 post-partum. When results for all 90 mothers were pooled, regardless of gestational age, insulin concentration in breast milk fell significantly from day 3 to day 10 (50.1 (34.6) v 41.1 (28.5) µU/ml; p = 0.01; mean (SD)). However, this decrease was only significant for mothers delivering at term (37–41 weeks).

Shulman conducted the first clinical trial giving insulin (4U/kg/day) per os for preterm infants up to 28 days after delivery. They achieved full enteral feeding within 11 days – comparedto 20 days in the control group.  Lactase activity in treated infants was twice of that in untreated patients (p=0.02) and the gastric residuals were 40% less than in controls (p<0.001)

(Shulman RJ:

Effect of enteral administration of insulin on intestinal development and feeding tolerance in preterm infants: a pilot study

Archives of Disease in Childhood - Fetal and Neonatal Edition 2002; 86: F131-F133.)

Future studies will need to address the supplementation of insulin in term infants and assess the efficacy of such supplementation in enhancing gut maturation and prevention of later noncommunicable diseases such as allergy, autoimmune diseases and obesity.

(Raanan Shamir, Naim Shehadeh

Insulin in human milk and the use of hormones in infant formulas

Nestlé Nutr Inst Workshop Ser

2013;77:57-64. doi: 10.1159/000351384. Epub 2013 Aug 29.)

Recent data suggest that BM insulin decreases over the first month of lactation, and significantly decreased by HoP between 13–46%.

The postnatal changes in human milk insulin concentrations and the effects of oral insulin supplementation on the immature intestinal mucosa warrant further investigations.

  1. A decade ago, several papers highlighted the presence of bacteria in sites once considered sterile (placenta, amniotic fluid, meconium). An article published in “Nature” in 2018 analyzes the history of controversies on this issue. In the present paper these theories are not discussed in detail.

The Human Microbiome Project provides an opportunity on the characterization, physiology and significance of the microbiota in multiple compartments of the body.

Aagaard et al compared the placental microbiota discovering that the main similarities were found with the oral microbiota: the hypotheses is that bacteria can reach the placenta by a hematic route.

(Aagaard, K.; Ma, J.; Antony, K.M.; Ganu, R.; Petrosino, J.; Versalovic, J. The placenta harbors a unique microbiome. Sci. Transl.

Med. 2014, 6, 237ra65.)

In the recent years, several studies conducted on meconium support the view that the fetus is exposed to microbiota in utero though previously the fetus, amniotic fluid, and placenta had been thought to be sterile.

In summary, this is an excellent article. More figures and tables might help to understand the text better. I also suggest that the above-mentioned literature data - at least some of them - should be taken into consideration.

Author Response

Reviewer #4

Flavia Indrio et al sent an article entitled „DEVELOPMENT OF THE GASTROINTESTINAL TRACT IN NEWBORN: CHALLENGE FOR AN APPROPRIATE NUTRITION” for publication to Nutrients.

This paper provides an overview on the maturational features of the main gastrointestinal tract (GI) functions, their implications following premature delivery focusing on the developmental differences in intestinal digestion and absorption functionality, motility, gut-brain axis interaction and microbiome.

Adequate nutrition of preterm infants still remains a challenge with regard to achieving a sufficient amount of bioactive substances assuring linear growth and neurocognitive development. During fetal life from the maternal circulation and amniotic fluid hormones are available until the development of fetal endocrine system will be completed. Preterm birth disrupts the maternal origin, the breast milk (BM) is the only source of maternal hormones. Feeding preterm infants, the own mother’s milk is the first choice, if it is not available, donor milk (DM) is recommended. It is important to note that milk composition of mothers who gave birth to preterm infants differs from breast milk produced for term infants. It was proven also that human milk reduces the incidence of necrotizing enterocolitis and several morbidities compared to formula fed infants. For microbiological safety DM milk is exposed to Holder pasteurization (HoP), however, this procedure may influence the levels of bioactive compounds in human milk.

Suggestions:

  1. In the present paper the effect of hormones and bioactive properties on anatomical and functional development of the GI tract is not emphasized to the extent of its importance.

Insulin plays an important role in intestinal maturation, also improves feed tolerance and has an impact on microbiome development.

Amniotic fluid contains carbohydrates, protein, fat, electrolytes, immunoglobulins, and growth factors. Insulin is present in the amniotic fluid (up to 20 μU/ml) and after skin keratinization completed (around 26th gestational week), amniotic fluid is the main source of insulin exposure to the GI tract and plays a role in development. Preterm birth abruptly interrupts these important fetal life processes and the local insulin exposure of the GI tract with detrimental consequences.

(J Neu, Nan Li: The Neonatal Gastrointestinal Tract: Developmental Anatomy, Physiology, and Clinical Implications Neoreviews (2003) 4 (1): e7–e13. https://doi.org/10.1542/neo.4-1-e7.)

Shehadeh N et al showed that the insulin concentration of human milk after full term delivery (60.2 (41.1) mU/ml) is significantly higher than that of cow’s milk (16.3 (6.0) mU/ml), and that insulin is barely detectable in infant formulas. (N Shehadeh, E Khaesh-Goldberg, R Shamir, R Perlman, P Sujov, A Tamir, I R Makhoul: Insulin in human milk: postpartum changes and effect of gestational age Arch Dis Child Fetal Neonatal Ed 2003;88:F214–F216)

Human milk insulin concentrations in mothers of preterm infants were not significantly different from those of full-term infants, on either day 3 or day 10 post-partum. When results for all 90 mothers were pooled, regardless of gestational age, insulin concentration in breast milk fell significantly from day 3 to day 10 (50.1 (34.6) v 41.1 (28.5) μU/ml; p = 0.01; mean (SD)). However, this decrease was only significant for mothers delivering at term (37–41 weeks).

Shulman conducted the first clinical trial giving insulin (4U/kg/day) per os for preterm infants up to 28 days after delivery. They achieved full enteral feeding within 11 days – comparedto 20 days in the control group. Lactase activity in treated infants was twice of that in untreated patients (p=0.02) and the gastric residuals were 40% less than in controls (p<0.001)

(Shulman RJ: Effect of enteral administration of insulin on intestinal development and feeding tolerance in preterm infants: a pilot study Archives of Disease in Childhood - Fetal and Neonatal Edition 2002; 86: F131-F133.)

Future studies will need to address the supplementation of insulin in term infants and assess the efficacy of such supplementation in enhancing gut maturation and prevention of later noncommunicable diseases such as allergy, autoimmune diseases and obesity.

(Raanan Shamir, Naim Shehadeh Insulin in human milk and the use of hormones in infant formulas Nestlé Nutr Inst Workshop Ser 2013;77:57-64. doi: 10.1159/000351384. Epub 2013 Aug 29.)

Recent data suggest that BM insulin decreases over the first month of lactation, and significantly decreased by HoP between 13–46%.

The postnatal changes in human milk insulin concentrations and the effects of oral insulin supplementation on the immature intestinal mucosa warrant further investigations.

We have now included the following mention to the important role of insulin in the digestive processes and gut motility including some relevant references among those suggested by the reviewer: “Insulin plays an important role in intestinal maturation and may also contribute to improve lactase activity following preterm birth. Amniotic fluid contains carbohydrates, protein, fat, electrolytes, immunoglobulins, and growth factors. Insulin is present in the amniotic fluid (up to 20 μU/ml) and after skin keratinization completed (around 26 weeks’ gestation), amniotic fluid is the main source of insulin exposure to the GI tract and plays a role in development. Preterm birth abruptly interrupts these important fetal life processes and the local insulin exposure of the GI tract with detrimental consequences (J Neu, Nan Li: The Neonatal Gastrointestinal Tract: Developmental Anatomy, Physiology, and Clinical Implications Neoreviews (2003) 4 (1): e7–e13. https://doi.org/10.1542/neo.4-1-e7.) Oral insulin administration in preterm infants up to 28 days after delivery has proved effective in increasing of two folds the lactase activity (Shulman RJ: Effect of enteral administration of insulin on intestinal development and feeding tolerance in preterm infants: a pilot study Archives of Disease in Childhood - Fetal and Neonatal Edition 2002; 86: F131-F133.)“

  1. A decade ago, several papers highlighted the presence of bacteria in sites once considered sterile (placenta, amniotic fluid, meconium). An article published in “Nature” in 2018 analyzes the history of controversies on this issue. In the present paper these theories are not discussed in detail.

The Human Microbiome Project provides an opportunity on the characterization, physiology and significance of the microbiota in multiple compartments of the body.

Aagaard et al compared the placental microbiota discovering that the main similarities were found with the oral microbiota: the hypotheses is that bacteria can reach the placenta by a hematic route.

(Aagaard, K.; Ma, J.; Antony, K.M.; Ganu, R.; Petrosino, J.; Versalovic, J. The placenta harbors a unique microbiome. Sci. Transl. Med. 2014, 6, 237ra65.)

In the recent years, several studies conducted on meconium support the view that the fetus is exposed to microbiota in utero though previously the fetus, amniotic fluid, and placenta had been thought to be sterile.

Following this reviewer’s comment, we have briefly mentioned this important highlight on the presence of bacteria in sites once considered sterile hypothesized following meconium-based studies.